



# Causes, consequences and implications of the 2023 landslide-induced Lake Rasac GLOF, Cordillera Huayhuash, Peru

Adam Emmer[1], Oscar Vilca[2], Cesar Salazar Checa[3], Sihan Li[4], Simon Cook[5], Elena Pummer[6], Jan Hrebrina[6], and Wilfried Haeberli[7]

[1]University of Graz, Austria
[2]Instituto Nacional de Investigación en Glaciares y Ecosistemas de Montaña (INAIGEM), Peru
[3]Autoridad Nacional del Agua (ANA), Peru;
[4]University of Sheffield, UK
[5]University of Dundee, UK
[6]Norwegian University of Science and Technology (NTNU), Norway
[7]University of Zurich, Switzerland

*Correspondence to*: Adam Emmer (adam.emmer@uni-graz.at OR aemmer@seznam.cz)

**Abstract.** Glacierized Peruvian mountain ranges are experiencing accelerated, climate change-driven glacier ice loss. Peru's second highest mountain range, the Cordillera Huayhuash, has lost about 40% (~34 km$^2$) of its glacier cover since the 1970s. Newly exposed landscapes are prone to a number of hazard processes including the formation and evolution of glacial lakes, changing stability conditions of mountain slopes, and rapid mass movements. In this study, we integrate the analysis of meteorological data, remotely sensed images and field observations in order to document the most recent (February 2023) large mass movement-induced glacial lake outburst flood (GLOF) from moraine-dammed Lake Rasac. The GLOF was triggered by a mass movement from the failure of an arête ridge with an estimated volume of 1.1 to 1.5 x 10$^6$ m$^3$; this occurred from the frozen rock zone with cold, deep permafrost, and was preceded by several small-magnitude precursory rockfall events. The reduced stability of the frozen rocks in the detachment zone most likely relates to deep warming, but not to especially critical conditions of warm permafrost with higher amounts of unfrozen water. Further, we describe the surprisingly short-distanced process chain (attenuated by the Lake Gochacotan located 3.5 km downstream from the detachment zone) and analyze the transport of large boulders with the use of hydrodynamic modelling, revealing that flow velocities > 5 m/s must have been reached in the case of translational motion, and > 10 m/s in the case of rotational motion of the largest transported boulders (diameter > 3.5 m). In addition, we analyze climate trends over past seven decades as well as meteorological conditions prior to the GLOF, revealing a statistically significant atmospheric temperature rise and thermal anomaly before the event. Climate change effects (warming air and permafrost temperatures) served to hasten the failure of an already critical geological situation. This study helps us to understand (i) mechanisms, amplification and attenuation elements in GLOF process chains; and (ii) altering frequency-magnitude relationships of extreme geomorphic processes in rapidly changing high-mountain environments on a regional scale (both large magnitude rockfalls and GLOFs). This study supports earlier work that indicated an increasing frequency of large mass movement-induced GLOFs originating from ice-related effects of glacial de-buttressing and warming permafrost in recent decades.



## 1 Introduction

The world's high-mountain regions are losing glacier ice and permafrost at an accelerated pace in response to climate change (Hugonnet et al., 2021), with important implications for downstream areas (Clague et al., 2012; Knight and Harrison, 2013; Huss et al., 2017; Immerzeel et al., 2020). There are growing concerns about potentially disastrous, far-reaching process chains originating from a warming cryosphere (Haeberli et al., 2017; Ding et al., 2021), as exemplified by recent disasters such as the 2020 Lake Jinwuco outburst (Zheng et al., 2021), the 2021 rock and ice avalanche at Chamoli in India (Shugar et al., 2021), the 2022 Marmolada glacier collapse in Italy (Olivieri and Bettanini, 2023), or the 2023 Lake South Lhonak cross-border outburst impacting vast areas along the Teesta River in India and Bangladesh (SANDRP, 2023). Among the hazard processes of greatest concern for mountain communities and piedmont areas are glacial lake outburst floods (GLOFs) – sudden releases of water from glacial lakes (O'Connor et al., 2013; Emmer et al., 2021; Zhang et al., 2024).

The Peruvian Andes have witnessed >100 GLOFs in the past century, some of which have had catastrophic consequences (Carey, 2005; Emmer et al., 2022b). Strikingly, the two cordilleras located in Northern Peru (C. Blanca and C. Huayhuash) exhibit a higher concentration of major GLOFs, including the 1941 Lake Palcacocha disaster that destroyed part of Huaráz (Huggel et al., 2020; Wegner, 2024). Emmer et al. (2020) showed that, while major moraine dam failure-induced GLOFs in the late 1930s to early 1950s were predominantly triggered by ice avalanches and calving processes, more recent events were more frequently triggered by landslides and rockfalls. Despite recent advances in compiling GLOF inventories (Emmer et al., 2022a; Veh et al., 2022; Lützow et al., 2023), forecasting and predicting GLOF occurrence in space and time is challenging due to: (i) a wide range of possible combinations of lake and dam characteristics favoring GLOF occurrence, triggers and mechanisms (Emmer and Cochachin, 2013); (ii) the often non-periodic, one-off nature of GLOFs (especially for GLOFs originating from bedrock- and moraine-dammed lakes; Lützow et al., 2023); as well as (iii) dynamic conditions, including the evolution of proglacial lakes driven by glacier retreat (Shugar et al., 2020), and changing slope stability conditions driven by glacier recession and permafrost warming (Stoffel and Huggel, 2012; Haeberli et al., 2017). Therefore, the frequency-magnitude relationship and derived recurrence interval approaches traditionally used in flood hazard assessments on a catchment scale cannot be applied to GLOFs; the estimates of GLOF probabilities are instead derived from detailed understanding of past events at a regional scale, with consideration for how future conditions will evolve (Haeberli et al., 2022; Emmer, 2024).

This study aims to examine the preconditions, trigger and impacts of the most recent large ice-rock failure-triggered 2023 Lake Rasac GLOF (Cordillera Huayhuash) and to discuss whether and, if so, then how, this event can be attributed to climate change. The importance of documenting such events lies in better, evidence-driven understanding of: (i) mechanisms, amplification and attenuation elements in GLOF process chains; and (ii) altering frequency-magnitude relationships of extreme processes in rapidly changing high-mountain environments on a regional scale (both high-magnitude rockfalls and GLOFs).



## 2 Study area and previous lake and GLOF research

### 2.1 The Cordillera Huayhuash

The Cordillera Huayhuash (76.8-77.0°W 10.0-10.5°S, Fig. 1), located 200 km north from Lima, is the second highest
mountain range in Peru (Nevado Yerupajá, 6,617 m asl), covering an area of nearly 1,200 km2 (INAIGEM, 2018). The
geological setting of Cordillera Huayhuash is complex and comprises faulted and folded Cretaceous sedimentary carbonates
of marine origin, Tertiary silicic volcanic rocks and intrusive rocks (tonalite, granite, granodiorite, diorite), in places covered
by Quaternary deposits of glacial and glacifluvial origin, alluvium and colluvium. The main fault zone is oriented in a North-
South direction and is parallel to the main massif of the Cordillera Huayhuash (IGM, 1975; Hall et al., 2009).



**Figure 1: Location of the Cordillera Huayhuash in Peru (A); lakes (Bat'ka et al., 2020), glaciers (RGI Consortium, 2017) and topography (ASF, 2024) of the Cordillera Huayhuash (B) and detail of upper Jahuacocha and Rasac valleys (C).**



The high-elevation areas of the Cordillera Huayhuash (above approximately 4,500 - 5,000 m asl) are glacierized and exhibit several generations of moraines from past glaciations (Hall et al., 2009). The last significant glacier advance occurred during the Little Ice Age, which subsided in the second half of the 19th Century. According to INAIGEM (2018), glacier extent decreased from 86.89 km$^2$ in 1975 to 53.06 km$^2$ in 2016. McFadden et al. (2011) estimated an average snowline altitude (an approximation of equilibrium altitude) of 5051 m asl in 1986 – 2005. Repeated Holocene advance-retreat cycles

of glaciers preconditioned the formation of different types of glacial lakes. Using Landsat imagery, Wood et al. (2021) mapped 129 lakes with total area of 6.92 km$^2$, within 1km buffer from the glaciers, while Bat'ka et al. (2020) used high-resolution images from Google Earth collection to map 270 lakes with total area of 6.32 km$^2$ in the whole Cordillera Huayhuash. Two of these lakes – Rasac (76°56'18''W, 10°15'50''S; 4654 m asl) and Gochacotan (76°56'25''W, 10°14'58''S; 4,285 m asl) – are the focus of this study (see Section 2.2).

High-mountain lakes can be characterized by short longevity (Costa and Schuster, 1988; Korup and Tweed, 2007) and several lake outburst floods were documented from the Cordillera Huayhuash previously, some of which with major geomorphic and societal impacts. Emmer et al. (2022), in their Peru-wide GLOF inventory, listed 14 GLOFs that occurred there (compiled and updated from: Kinzl et al., 1954; Zapata, 2002; Bat'ka et al., 2020). The most recent event – the 2023 GLOF from Lake Rasac – is reported and characterized in this study.

## 2.2 Lake Rasac

Lake Rasac (Figs. 1 and 2) was a moraine-dammed water body located at 4,654 masl behind the moraine deposited by Rasac glacier in the north-facing Rasac valley, a left bank tributary of Jahuacocha/Achin stream (Salazar and Valverde, 2022). Considering the elevation of this moraine and its position in the valley system of moraine assemblages, it is assumed they have been last reached during the Little Ice Age advance. While the lake did not have surface outflow (dam freeboard was 10

- 15 m, depending on lake water level) it was drained by a system of subsurface channels through the moraine dam and the water level (area and volume) fluctuated both inter- and intra-seasonally. The area of the lake varied between 53,000 and 64,000 m$^2$ (ANA, 2014, INAIGEM, 2018) and the depth was not known. Using the empirical equation developed by Muñoz et al. (2020), the estimated mean lake depth is 5 m and so the estimated lake volume is approximately 300,000 m$^3$.

      Bat'ka et al. (2020) conducted a comprehensive assessment of mountain range-wide GLOF susceptibility of 46

lakes each larger than 20,000 m$^2$. They found that for Lake Rasac, the most probable GLOF scenario involved dam overtopping as a result of a landslide into the lakeand ranked this lake as the 16th most susceptible from all assessed lakes in the C. Huayhuash (other moraine-dammed lakes reached higher scores because of their surface outflows; see the assessment methodology of Emmer and Vilímek, 2014).





Figure 2. Lake Rasac and its surroundings in July 2019. (A) upstream view towards the head of the Rasac valley (looking south);
(B) downstream view towards both lakes Rasac and Gochacotan (looking north) from near the Rasac Punta (5,129 m asl); (C)
oblique view towards the Rasac arête ridge (looking east) with the 2023 released zone highlighted. Images: AE.

## 3      Data and methods

### 3.1 Remotely sensed data and analysis

We integrated the analysis of satellite images and derived products (such as digital elevation models, DEM; see Tab. 1) from
various sources for event description and geomorphic interpretation. These data were used to: (i) estimate spatial
characteristics of mapped features from high-resolution optical images (e.g., area of the lake, areas of landslide release and
GLOF impact zones, location of largest transported boulders); (ii) derive topographical and morphometric characteristics
(e.g., a slope of the longitudinal profile along the process chain); (iii) narrow down the timing of the GLOF from images
with fine temporal resolution (Planet and Sentinel images). Further, we cross-checked the interpretation of remotely sensed
images against field observations (field visit in July 2019; post-event field images and testimony of an eyewitness – a



member of the community settled in the Rio Jahuacocha valley few km downstream). All the analyses of remotely sensed images were undertaken in open source QGIS software (QGIS, 2024).

**Table 1**. Remotely sensed data used in this study.

| Remote sensing data / products | Spatial resolution | Temporal coverage | Reference |
|---|---|---|---|
| Google Earth collection (Maxar Technologies, CNES / Airbus) | <0.5 m | 2010 – 2023 (7 high-resolution cloud-free scenes) | Google Earth Pro, 2024 |
| Planet Labs images | 2-3 m | 2016 – 2024 | Planet Team, 2024 |
| Sentinel collection | 15 m | 2016 – 2024 | EO Browser, 2024 |
| Landsat 8 | 15 m (panchromatic) | 1980s - 2024 | LandsatLook, 2024 |
| ALOS PALSAR DEM | 12.5 m | N/A (acquisition date 2011) | ASF, 2024 |

**3.2 Climate data and analysis**

**3.2.1 Climate trends**

Given that meteorological station records in Peru are very sparse and limited (none of which have consistent records going to 130  2023), we employed the state-of-the-art ERA5 gridded reanalysis products, ERA5-land (~9km horizontal resolution) and ERA5 (~31km horizontal resolution), from the European Centre for Medium-Range Weather Forecasts (Hersbach et al., 2020). We use total rainfall (liquid plus solid), snowfall, snowfall to total rainfall ratio, 2m temperature, land surface skin temperature, soil temperature in 4 layers (layer 1: 0-7cm; layer 2: 7-28cm; layer 3: 28-100cm; layer4: 100-289cm), 0°C level (from ERA5, defined as 'the height above the Earth's surface where the temperature passes from positive to negative 135  values'), surface sensible heat flux, and surface latent heat flux from ERA5-Land using a 1° by 1° box [-77.5, -76.5, -10.75, -9.75] covering Rasac to investigate the anomalous (compared with climatological mean state) land surface and near-surface atmospheric states associated with the event. The atmospheric circulation variables zonal (U) wind, meridional (V) wind, and geopotential height at different vertical levels (850hPa, 500hPa, and 250hPa, representing lower, middle, and upper troposphere) were also extracted from ERA5 covering region -85.0, -65.0, -20.0, 0.0), and are used to investigate any 140  possible anomalous circulation patterns associated with the event. It should be noted that the variables from ERA5 are not directly assimilated, but these are generated by atmospheric components of the Integrated Forecast System (IFS) modelling framework.



**3.2.2 Climate change attribution**

We used the meteorological variables averaged over the 1° by 1° box [-77.5, -76.5, -10.75, -9.75] covering Rasac, taking the
February 1-day maximum as the event definition. As outlined in Philip et al. (2020), we assess how the likelihood of
occurrence of the event as defined above has changed due to anthropogenic climate change. We use a Generalized Extreme
Value (GEV) distribution that scales with global mean surface temperature-GMST (i.e., using GMST as covariate, Philip et
al. 2020 and Van Oldenborgh et al. 2021). The resulting probability ratio (PR) for the event is calculated for the comparison
between the climate of 2023 and preindustrial climate- 1.2 degrees cooler than current climate. PR is calculated as the
probability of occurrence under the current climate, divided by the probability of occurrence under preindustrial climate.

We acknowledge that in typical statistical event attribution studies, climate model outputs are also analysed (often
take from the most up to date Coupled-Model-Intercomparison-Project global earth system models, or regional model
experiments) through a model validation, multi-method attribution (combining observed and model results) to arrive at a
synthesized attribution statement. In this study, we did not perform analyses based on climate model outputs for two reasons:
(1) the Rasac 2023 GLOF is a localised event with no identifiable impact on humans and the built environments, and not
directly triggered by meteorological variables but triggered by mass movements (as detailed below); and (2) the attribution
of temperature anomalies is mainly to demonstrate the anthropogenic climate change impact on local warming associated
with the GLOF, rather than establishing a causal chain.

**3.3 Hydrodynamic analysis**

If the slope conditions and the composition of the terrain are known, and there is clear evidence of deposited boulders and
cobbles, the minimum water velocity that moved them can be estimated. Various empirical relationships exist for calculating
the critical flow velocity and shear stress required to initiate particle motion (Van Rijn, 2019). However, these formulae are
predominantly derived for fine spherical sediments or stones and under boundary conditions that do not align with the
impulse waves generated by landslides. To calculate boulder transport, we employ a method considering the balance of
forces and torque acting on boulders modeled as simplified rectangular shapes (Bressan et al., 2018). Since there is no
available survey detailing the size and shape of the boulders in the impact area for this analysis, we assume that the boulder's
dimensions are characterized by a height half the size of their width and length, which are considered equal. This assumption
aims to approximate the realistic shape of boulders where, typically, the boulder height does not match the width and length.
Like other sediment transport phenomena, the forces acting on the boulder include:


Gravity force:  $\quad G = gV(\rho_s - \rho_w V_w / V)$  $\qquad$ (0.1)

Drag force:  $\quad F_D = \dfrac{1}{2} A_f v^2 \rho_w C_d$  $\qquad$ (0.2)



Lift force:

$$F_L = \frac{1}{2} A_b v^2 \rho_w C_l \qquad (0.3)$$

Friction force:

$$F_S = \mu (G - L) \qquad (0.4)$$


Where $g$ stands for acceleration due to gravity (9.81 m/s²), $V$ is the volume of the boulder (m³), $V_w$ is the submerged volume based on the water depth $h$ (m), $\rho_s$ is boulder density (for granodiorite we assumed 2700 kg/m³), $\rho_w$ is water density (1000 m³/s), $A_f$ represents the front wetted area of the boulder (m²) based on the water depth $h$ (m), $A_b$ represents the bottom area of the boulder (m²), $v$ is the depth-averaged approaching speed velocity (m/s). The drag coefficient $C_d$ is assumed to be 2.05 for

a partially submerged boulder and 1.05 for a fully submerged boulder of cubic shape (Bressan et al., 2018). $C_l$ is the lift coefficient with a constant value of 0.178, and $\mu$ is the friction coefficient with a constant value of 0.5 (Nott, 2003; Lorang, 2011; Bressan et al., 2018). Current literature does not provide comprehensive guidelines for boulder transport. Friction, drag, and lift coefficients, as used in our study, are sourced from general literature (Cengel et al., 2013). Accurate determination of these coefficients would require specific experimental or mathematical models, which are beyond the scope

this study.

By applying a force balance in the direction of flow and a torque balance around the rear edge of the boulder, the required velocity $v$ (m/s) to disrupt this balance and initiate the incipient motion of the boulder can be calculated. It is assumed that if a velocity disrupts the force balance, translational motion begins, while if the velocity disrupts the torque balance, rotational motion initiates (Nandasena et al., 2013; Bressan et al., 2018). It is evident from the above discussion that

certain parameters are contingent on water depth. Due to the absence of data, a simplified relationship between wave depth and averaged velocity must be assumed.

### 3.4 Assessment of permafrost conditions

The assessment of permafrost conditions is based on the analysis of Google Earth Images, (sparse) climatic information, experience of borehole temperatures available in Europe, and scientific literature about slope stability of ice-clad and

perennially frozen mountain peaks. They constitute "best estimates" for a site without direct measurements, dealing with thermal conditions, subsurface and surface ice properties, and resulting hydro-mechanical aspects in view of climate-related stability changes.



## 4 Results: description and interpretation of the 2023 event

### 4.1 Surface and subsurface ice, prior rockfall activity and the 2023 arête ridge failure

#### 4.1.1 Surface and subsurface ice in relation with slope stability

The NW-SE oriented Rasac arête ridge reaches elevations exceeding 6,000 m asl and is partly glacier-covered. Surface ice on very steep slopes (generally > 45°) occurs in two main forms: ice aprons (ice faces) and hanging glaciers. Both forms, together with ice crests and small summit ice caps, are abundant in the region. Ice aprons or ice faces are thin (meters), smooth, near-static, cold ice covers frozen onto steep perennially frozen rock faces. Like perennial ice patches in less inclined topography, such ice covers cool the ground because they have a high albedo, and because they buffer against the penetration of above-zero temperatures whilst allowing penetration of cold waves to the ground below. The ice itself may be thousands of years old, and their disappearance due to atmospheric temperature rise induces freeze-thaw cycles within the uncovered frozen rocks, followed by minor to major rock destabilization (Ravanel et al. 2023). Hanging glaciers are much thicker and move from a less inclined accumulation area towards the front, where ablation tends to be by dry calving. Even with strongly negative mean annual air temperatures, they can be polythermal in that meltwater percolating from the surface and refreezing at depth of the firn constitutes a strong source of latent heat, inducing temperate and often temperate firn/ice with high water pressure (Margreth et al. 2017). Two important consequences of such conditions are that (i) the stability of such glaciers in steep terrain mainly relates to the cold, near-vertical ice front frozen to bedrock in a narrow zone at the lower margin, and (ii) that the temperate firn area with percolating meltwater can introduce strong local thermal anomalies (Haeberli et al. 1997) and complex thermo-hydraulic conditions in the underlying permafrozen rocks (Haeberli et al. 2004). Macroscopic subsurface ice in perennially frozen bedrock usually fills cracks and fissures. As the original formation of such ice involves efficient frost cracking and frost wedging, such ice fillings can be massive as sometimes visible in permafrost tunnels (Haeberli et al. 1979, Mamot et al. 2021) or as uncovered immediately after detachments of rock avalanches (Gruber and Haeberli 2007).

Under these conditions, mass movements can be released. One of the areas that developed frequent rockfall activity is located upstream from lake Rasac at an elevation between 5,500 and 5750 m asl (see Fig. 2C). The detachment zone of the 2023 event was covered by ice aprons rather than hanging glaciers. As a consequence, rather homogenous spatial patterns of subsurface temperatures can be expected. The existence of ice aprons on the WSW-oriented detachment side of the Rasac ridge and the lack of any ice cover on its other, ENE-oriented side is likely to enhance the temperature differences between the two sides and the related horizontal flux of heat across the ridge towards the detachment site.

#### 4.1.2 Prior rockfall activity and the 2023 GLOF-triggering failure

Slopes on the right bank of the Rasac lake are formed of tilted layers of sedimentary rocks inclined towards the Rasac valley (Fig. 3) that favor disintegration and that precondition mass movements. Available high-resolution satellite images with sub-metric resolution going back to 2010 (Fig. 3B to E) reveal that the release zone gradually migrated to higher elevations,



230    continuously undermining an overhanging block that eventually failed in February 2023. Small rockfalls from this release
       zone have occurred frequently since 2018, i.e., five years before the major failure, as documented by the Sentinel time series
       2017-2023 (Fig. 4); see also Fig. 2C) and the analysis of Planet images.

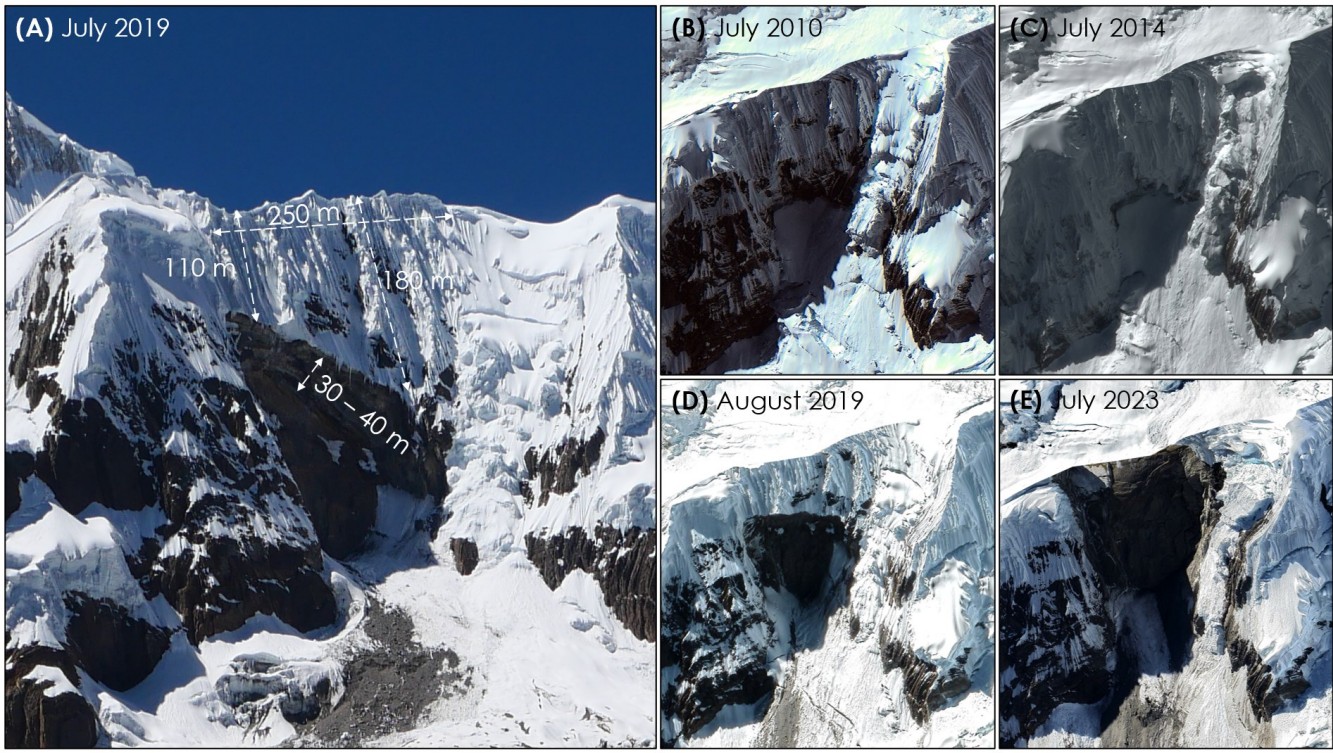

235    **Figure 3. Release zone of the GLOF-triggering landslide. (A) shows field image of hanging block as seen from the opposite slope in
       July 2019 (Image: AE); (B) to (E) show pseudo-3D view of release zone between 2010 and 2023 in © Google Earth (Maxar
       images, CNES / Airbus images). Note very little rockfall activity (change of a release zone) between 2010 and 2014.**

       The GLOF-triggering failure of an overhanging section of very steep (>60°) part of the Rasac arête ridge occurred
       around noon on 12th February 2023. The head scarp was located at an elevation between 5,750 and 5,800 m asl; the
240    approximated planar extent of the release zone is 250 m wide and 150 m high on average (Fig. 3A). Field images taken in
       2019 reveal that the thickness of the hanging block is approximately 30 to 40 m, resulting in a volume estimation of 1.1 to
       1.5 x $10^6$ m$^3$. This means that the landslide would be considered to be 'large' under the landslide classification scheme of
       McColl and Cook (2024). The 2023 landslide material was rock mixed with ice. In view of the thin (meters), co-detached ice
       aprons, the relative amount of ice in the 2023 rock/ice avalanche was probably on the order of 10 to 20%. The vertical drop
between the head scarp and Lake Rasac is 1,200 m (Fig. 5).



**Figure 4. Sentinel image time-series depicting rockfall activity and development of the release zone between 2017 and 2023. Note gradually developing head scarp and release zone (R) and increasing amount of material deposited on the glacier and slopes above the eastern bank of the lake (D).**

Large slope failures are often triggered by earthquakes or extreme hydro-meteorological conditions. According to the USGS earthquake catalogue (USGS, 2024), 71 earthquakes with Magnitude >4.5 occurred within the 200 km buffer distance from the landslide site between 2015 and 2024, with the highest magnitude reaching M5.6 (95.8 km deep near Lima in June 2022). However, if the buffer is decreased to 100 km, the search returns only 5 earthquakes (highest magnitude M4.9). No earthquake occurred within the 200 km buffer distance from the landslide site near the time of its occurrence and so we exclude the possibility of direct earthquake triggering as well as preconditioning. Instead, we examine climate change preconditions and evaluate the role of subsurface ice and degrading permafrost (see Section 4.2).



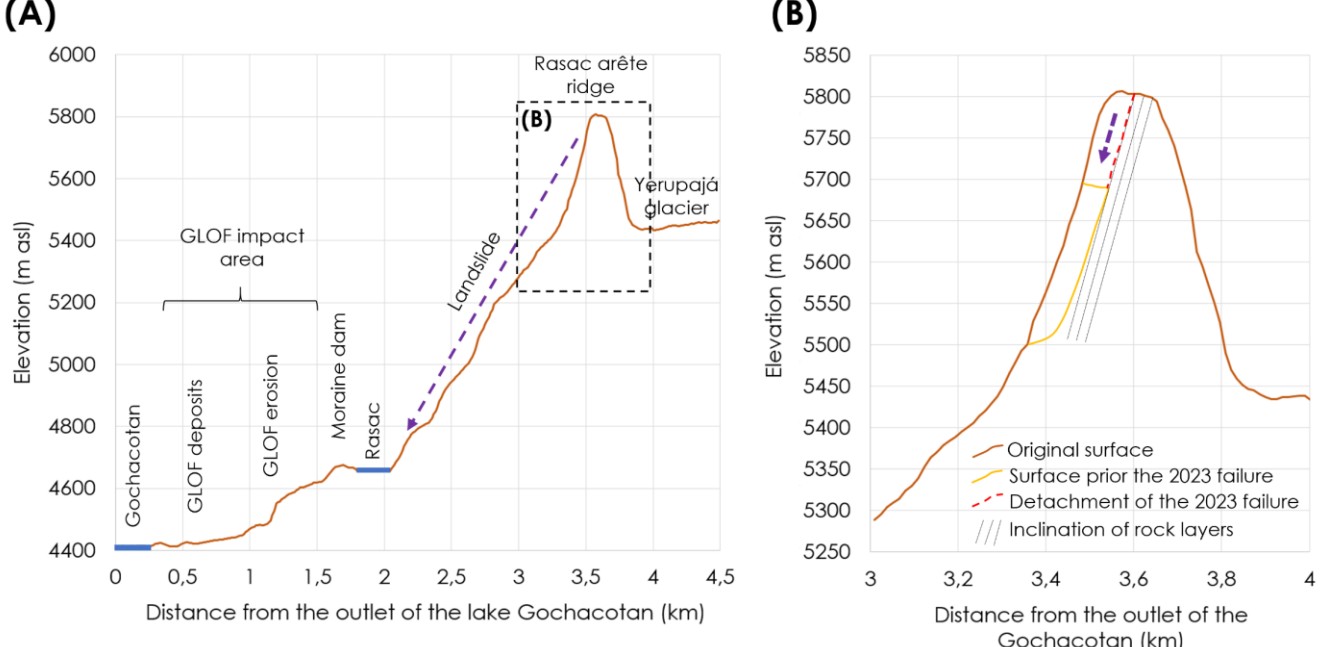

**Figure 5. Topographical profile of the whole process chain (A) and schematic sketch of the 2023 Rasac arête ridge failure (B). Note the x:y ratio 2:1. DEM: 12.5m ALOS PALSAR (ASF, 2024).**


## 4.2 Climate change, thermal and permafrost conditions

### 4.2.1 Changing climate in the study area between 1950 and 2023

From a longer-term climate change perspective over several decades, the study region has been warming significantly, receiving less snowfall, and experiencing a rise in the 0°C elevation. Over the studied region, all temperature metrics are

showing substantial increase over the past 74 years up to 2023 (Fig. 6, Supplementary Figs. S1 to S8). February 2023 was warmer compared to the historical mean, but does not stand out as anomalously warm compared to other individual years (Fig. 6A). Although there is no significant trend in total rainfall (Fig. 6D), solid rainfall, i.e., snowfall in February and January, has been seeing a consistent decrease, with February experiencing a stronger decrease (Fig. S1). As a result, the ratio of snowfall to rainfall has also been decreasing over the study region in January and February. Interestingly, but not

surprisingly, the 0°C elevation has also been rising significantly in January and February with February experiencing a stronger increase (Figs. 6E and 6F).





**Figure 6. Selected climate variables, averaged over the 1° by 1° box [-77.5, -76.5, -10.75, -9.75] covering Rasac, showing their seasonal cycles and historical trends. Seasonal cycle of 2-m air temperature (°C) for 1950-2023 shown in panel (A), and the historical trend for February and January shown in panel (B). Seasonal cycle of total precipitation (rainfall plus snowfall, in mm/day) for 1950-2023 shown in panel (C), and the historical trend for February and January shown in panel (D). Seasonal cycle of 0°C level (m) for 1950-2023 shown in panel (E), and the historical trend for Feb and Jan shown in panel (F).**



### 4.2.2 Thermal conditions, the role of permafrost and hydro-mechanical aspects

Ice-filled clefts and negative temperatures have strong hydro-mechanical effects on the stability of steep icy peaks. The ice-
filling of cracks and fissures reduces the hydraulic permeability of the frozen rock mass to essentially zero. New cracks
without ice may, however, form through slow rock deformation. Where water enters these open cracks, efficient advective
heat transport, local rock wedging and high water pressures can result (cf. Hasler et al. 2012, Walter et al. 2019). The
strength of negative-temperature rocks with ice-filled clefts is high but decreases with increasing temperature (Fig. 7).
Critical conditions develop especially in "warm" permafrost with increasing amounts of unfrozen water at temperatures
between about -1.5 and 0°C. This is due to the fact that not only do the rock and ice components lose strength, but so do the
rock-rock and rock-ice contacts, where temperatures and unfrozen water contents increase (Krautblatter et al. 2013).
Complete permafrost thawing with elimination of ice then leaves to a slight recovery of stability but still at a much lower
level than with cold permafrost (Fig. 7).

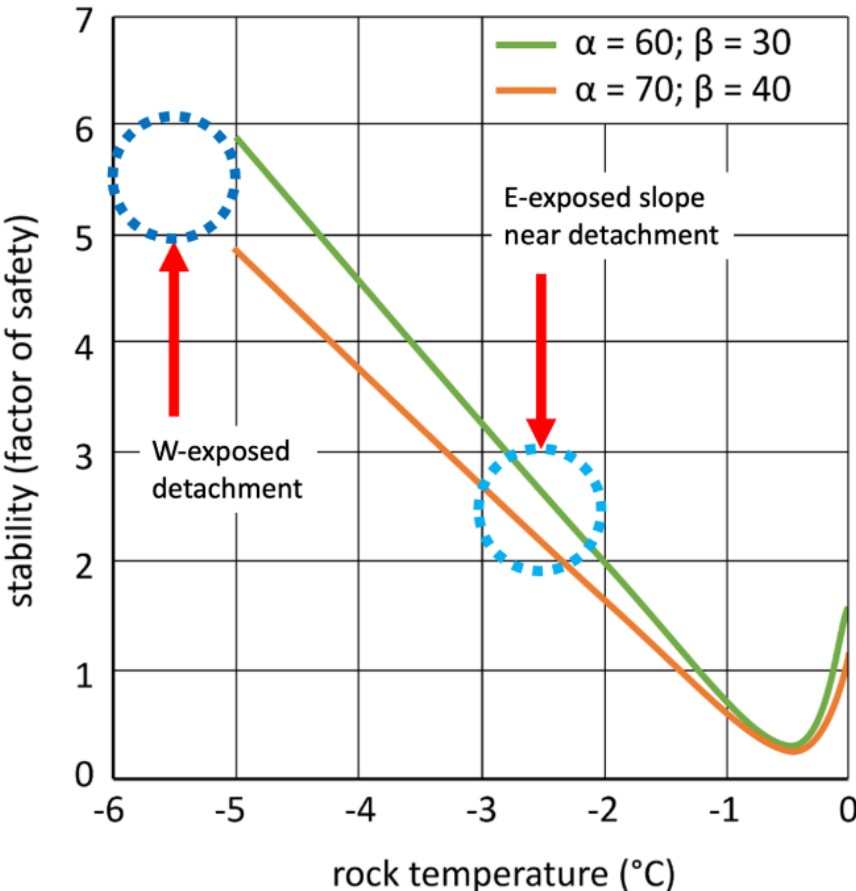

**Figure 7. Stability of frozen rocks with ice-filled clefts as a function of rock temperature from centrifuge experiments for two sets
of slope angle α and inclination of discontinuity β (redrawn from: Davies et al. 2001). Red arrows point to best guesses of
permafrost temperatures at the Rasac detachment site.**



The detachment site is in a very steep, WSW-oriented slope at an altitude of 5,600 m asl. reaching up to the ridge top at 5,780 m asl. (Fig. 1C). As such steep faces cannot accumulate a thick, thermally insulating snow cover, their mean surface temperature is close to the mean annual air temperature. Schauwecker et al. (2017) estimate the altitude of zero mean annual air temperature in the region to be about 5,000 m asl. Applying an environmental lapse rate of 6 °± 2°C/km (Steward-Jones and Gruber 2023) provides a rough estimate of the mean annual air and surface temperature at the altitude of the

detachment site of around -4 ± 2°C. This indicates cold and deep permafrost.

        The N – S oriented Rasac ridge induces pronounced thermal contrasts between its E- and W-facing slopes. Due to atmospheric humidity on average increasing during the day as a consequence of convection/cloud formation, E-oriented slopes, which receive morning sun, tend to be considerably warmer than W-exposed slopes receiving evening irradiation reduced by increased atmospheric humidity/clouds. This effect is independent of latitude. In the Alps, the difference between

permafrost occurrence on E- and W-oriented slopes is around 500 m (Haeberli 1975, Keller 1992). This translates into a temperature difference between the two orientations of about 3°C. Correspondingly adjusting the estimated mean surface temperature for W-exposure by -1.5°C leads to a best estimate of about -5.5 ± 2°C at the W-oriented detachment site (Fig. 7). Applying a corresponding adjustment of + 1.5°C to the E-oriented slope of the ridge results in a mean surface temperature at the same altitude of -2.5 ± 2°C. It follows that the temperature pattern is strongly asymmetric. The isotherms

between the two sides with a temperature difference of about 3°C must be near-vertical, causing heat flow inside the ridge to have a correspondingly strong horizontal component across the ridge from the warmer E- to the colder W-oriented face (cf. Noetzli et al. 2007). Such thermal asymmetries are not uncommon in detachment zones with permafrost of rock avalanches: Cathala et al (2024) provide a detailed analysis of slightly warmer but comparable NW-SE asymmetric thermal conditions at the detachment site of the Étache rock avalanche in the French Alps, using in situ measurements and numerical model

calculations. In the uppermost part of the Rasac detachment, where the ridge width is around 100 – 200 m, temperature gradients (around 1 - 3°C/100 m) indicate a horizontal heat flow component, which is comparable to the geothermal heat flow in flat topography under conditions of thermal equilibrium.

        Due to atmospheric warming since the Little Ice Age, subsurface temperature gradients at depth, as documented by thermal monitoring of deep mid- and high-latitude boreholes, are generally reduced within roughly the uppermost 100 m

below surface (Etzelmüller et al. 2020). The observation by Thompson et al. (1995), that a coldest temperature of -5.2°C in their borehole on the cold firn/ice saddle of Huascarán (6048 m a.s.l.) was measured at the ice/rock interface at 82.5 m depth, is in general agreement with the above-estimated surface temperatures and the warming-induced reduction of subsurface temperature gradients and related (vertical) heat flow as documented in mountain borehole temperatures at higher latitudes (Etzelmüller et al. 2020). The uppermost part of the Rasac ridge with the detachment site is most probably frozen

throughout. Subsurface heating due to ongoing atmospheric temperature rise is from both sides of the ridge. Only small remaining parts unaffected by colder historical/Holocene times may still exist deep inside the mountain (cf. transient 3D model calculations by Noetzli and Gruber 2009).



### 4.2.3 Meteorological conditions before the GLOF

To investigate whether anomalous meteorological conditions contributed to triggering of the event on 12th of February, we investigated temperature and precipitation metrics over the study region. The most salient feature is that almost all temperature metrics investigated here (apart from deep soil layer temperature; see Supplement), started to become anomalously warmer compared to the long-term mean (Fig. 8).

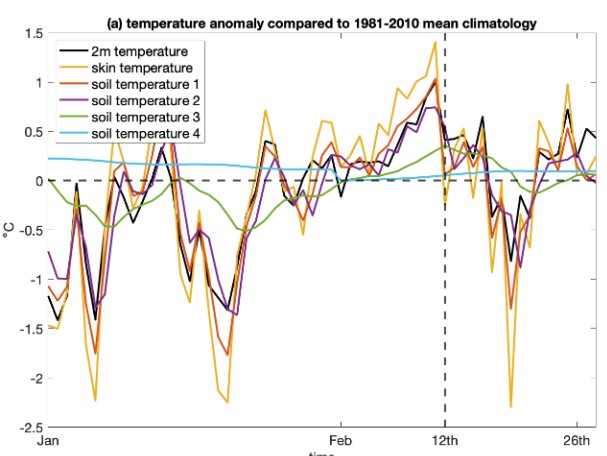
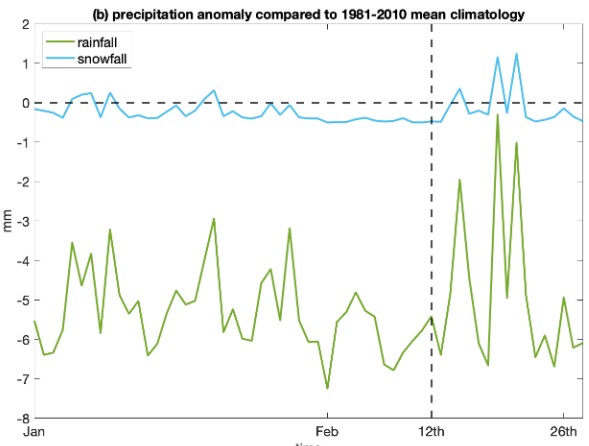


**Figure 8. Temperature (a) and precipitation (b) anomaly in January and February of 2023, compared to the 1981-2010 climatological mean. Dashed vertical line indicates the timing of the Rasac landslide and GLOF event on 12th February 2023.**

### 4.3 GLOF process chain, hydrodynamics and transport of large boulders

### 4.3.1 GLOF process chain

The 2023 rockfall into Lake Rasac created a displacement wave that overtopped the crest of the lake's moraine dam at two locations and impacted the section of valley between Lake Rasac and Lake Gochacotan (Fig. 9B). The impact area between the crest of the dam of the GLOF-producing Rasac lake and the downstream lake Gochacotan (Fig. 9C) covers 142,200 m$^2$. The reach of geomorphologically effective GLOF impacts is approximately 1.4 km with a vertical drop of 220 m (average slope of the trajectory 10°). The impact area is characterized by the presence of several large boulders with diameters > 3m

(Fig. 9D). These boulders were transported by the GLOF from the dam for a distance of > 600 m. Further geomorphic impacts include the erosion of existing stream networks as well as the creation of new branches of the stream network. The rockfall also hit a small bedrock-dammed lake located east from the moraine dam of lake Rasac. High resolution Planet Labs images reveal minor geomorphic impacts on the east-facing side of the ridge (part of the released material was deposited there, i.e., in direction towards Yerupajá glacier). Interestingly, and fortunately, only very limited impacts are observed



further downstream from lake Gochacotan and the GLOF did not reach the community settled near the Jahuacocha lake (3
km downstream from Gochacotan; see also Section 5.1).

**Figure 9. Impact area of the 2023 process chain. (A) and (B) show the location before (8/2019) and after (2023); (C) shows disappeared lake and entire GLOF impact area and (D) shows detail of impact area where both erosional and depositional features associated with the 2023 GLOF are visible. B1 to B5 refer to large boulders analyzed in Section 4.3.2. Images: Maxar images, CNES / Airbus images available from © Google Earth collection.**



### 4.3.2 Transport of large boulders

Large boulders (diameter > 3m) were transported within the GLOF impact area, indicating that a transient, high-energy flow
mobilized them. The boulders probably originated from a moraine dam, and their transport was facilitated by the pronounced
steepness of the first section of the downstream face of the moraine after wave impact. There are no apparent indications
within the impact zone suggesting the occurrence of hyper-concentrated flows or debris flows, characterized by densities
exceeding 1800 kg/m³ (Costa, 1988). According to Google Earth Pro imagery (07/2023) and in-situ photographs of the post-
GLOF impact area, no substantial morphological changes are apparent associated with any such high-density flows. Such
flows typically induce substantial morphological alterations, resulting in the stripping of the entire surface layer of the
landscape (Jakob et al., 2005). Nevertheless, clear evidence of erosion and deposition processes in regions 2 and 3 (Fig. 7C)
indicate a possibility of stratified flows. Given the above, it is reasonable to classify this event as a regular flood with a
combination of water and sediment, displaying Newtonian fluid characteristics.

The minimum velocities required for translational and rotational motion and corresponding wave height are
illustrated in Fig 10. The results encompass a range of boulder dimensions (corresponding to width and length, respectively)
from 0.5 m to 6 m, with observed boulder dimension estimated from aerial imagery. Each curve represents a distinct region
within our study site. It is assumed that the boulders originate from the dam crest with zero slope. For the largest observed
boulder with a dimension of 5.6 m, the minimum velocity required for incipient translational motion is shown to be 8.8 m/s,
with a corresponding wave height of approximately 7.9 m (see Fig. 10). Additionally, the minimum velocity for incipient
rotational motion is determined to be 14.6 m/s, resulting from an approximate wave height of 21.6 m.

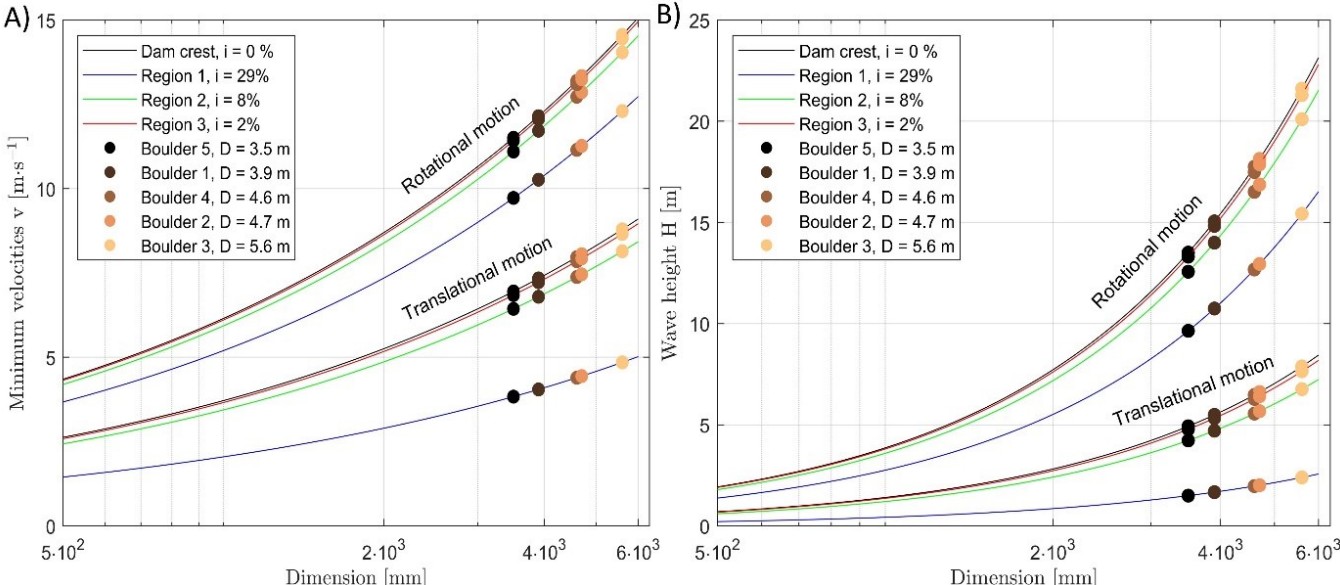

**Figure 10. Flow velocities needed to transport mapped large boulders.**





## 5 Discussion

### 5.1 The limited GLOF impacts


The 2023 Rasac GLOF is characterized by limited reach and impacts. This observation is in contrast with the 2020 Salkantaycocha GLOF that was induced by an ice-/rock-avalanche of similar volume and characteristics (estimated volume $1-2\cdot10^6$ m$^3$, vertical drop 1,000 m), but the reach of impacts exceeded tens of km (Vilca et al., 2021). This could be explained by: (i) dissipation of triggering rockfall energy on the bedrock slope before reaching Rasac lake; (ii) possible gradual (not

one-off) failure of the overhanging block (this may explain the two dam overtopping locations); (iii) the impact of a rockfall at the frontal part of the lake, diverting the primary direction of displacement wave towards the rear part of the lake; (iv) relatively small lake volume ($0.3\cdot10^6$ m$^3$ in Rasac vs. 0.8 million m$^3$ in Salkantaycocha); (v) attenuation role of Gochacotan lake.

The Sentinel and Landsat time-series reveal that lake Rasac persisted in a more limited extent after the GLOF. The

19th March 2023 scene (i.e., 4 weeks after the GLOF) shows a clear water body in the frontal part of the former lake basin, while no lake is identifiable on the next cloud-free image taken on 23rd March 2023. Such post-GLOF lake emptying is noticeably similar to what was observed in the case of the 2020 Salkantaycocha GLOF in Cordillera Vilcabamba (Vilca et al., 2021). The reasons for this quite unusual post-GLOF complete lake emptying may include: (i) enlargement of underground outflow channels caused by increased hydrostatic pressure immediately after the GLOF; (ii) opening of new

underground channels associated with erosion of distal face of moraine dam; (iii) decreased water inflow into the lake resulting from landslide deposition in its headwater area. Considering deposition of rockfall material in frontal part of the lake, option (iii) is rather unlikely.

### 5.2 The role of climate change and meteorological conditions in Rasac slope failure

Both the long-term change and the anomalous conditions prior to the event, in particular the deeper soil layer temperature

change suggest the role of permafrost warming in this event (see Section 4.2.3). However, the cold temperatures of the permafrost at the detachment site of the 2023 Rasac event make permafrost thawing to be unlikely as an important factor. The detachment must have taken place within the deeply frozen rocks. A general stability reduction due to deep permafrost warming, however, must be assumed to have played a partial role. Partial in the sense that the geological conditions must already have been in a critical state. Furthermore, penetration of some water from the warmer, E-oriented slope towards the

detachment cannot be excluded. Numerous small magnitude precursory events (see Sections 4.1 and 4.2) also initiated from cold and deep permafrost but eliminated the cooling ice aprons, which must have caused additional local warming and frost wedging. These precursory events must have led to the final destabilization of the main 2023 event despite the fact that this uppermost part of the ridge face remained covered by thin but thermally protective ice aprons.





### 5.3 Challenges in attributing the 2023 event to climate change

There are several challenges in attributing a single GLOF event such as the 2023 one to climate change. First, typically the initial step in a probabilistic event attribution study is to define the event in question. The central aim is to link the event definition as closely as possible to the impacts of the event, and quite often the event definition is focused on impacts on humans and/or the built environment. But in this case, the actual impact is mostly felt downstream and with no identifiable 'traditionally-defined' impacts. Second, event attribution studies are mostly focused on key contributing meteorological variables (e.g., temperature for heatwaves; heavy precipitation for flooding; heat, drought, and winds for fires). However, as demonstrated earlier, this sudden 2023 event was triggered by a mass movement, preceded by a few small rockfalls, instead of meteorological triggers. Although local temperature had been anomalously warm leading up the GLOF event, it was not the main trigger. Third, isolating the exact role of anthropogenic climate change for the 2023 event is challenging due to the various timescales of concern here. The anthropogenic influence on climate is undeniable when looking at decadal and centennial timescales given the human-induced increase in the energy content of the climate system. Glacier retreat, lake formation, and permafrost degradation are all clear consequences of a warming planet. These long-term changes increase the overall risk of GLOFs. However, for a single and sudden event like Rasac 2023 – with complex cumulative developments and effects – a complex system reaction influenced by ongoing anthropogenic climate change, it is challenging to conduct a meteorological variable-based attribution. Rasac 2023 is a case of climate change interacting with various natural processes (rockfalls, mass movements etc.) creating the condition for a GLOF.

A detailed and systematic attribution of glacial lake expansion and GLOFs to global warming has not yet been undertaken and the roadblocks to such efforts have been a lack of clear understanding of glacial lake evolution processes and a lack of modelling of such processes. Specific attribution of a single event is challenging, given the complexity of climate, geographic, and topographic factors acting on a regional level. A promising way forward to attributing GLOFs is to use integrated process chain glacier-climate modelling to address if and by how much glacial lake expansion and GLOFs are attributable to past total global warming and to the anthropogenic component. Nonetheless, while a single GLOF event might be hard to attribute, the overall trend of increasing GLOF risk due to climate change is clear, with ongoing climate change leading to glacier vanishing, lake formation and expansion, deep permafrost warming and degradation, and the resulting reduction in slope stability through glacial de-buttressing and reduced strength of frozen rocks.

With that being said, we still perform a simple observational-based analysis, attributing the anomalous temperature leading up to the event (which is the most salient feature through the climate section analysis), averaged over the 1° by 1° box [-77.5, -76.5, -10.75, -9.75] covering Rasac, and perform a simple observational-based analysis. Since the anomalous temperature was seen in a few different variables (Fig. 8a), and to make sure the results are robust, the attribution analysis was undertaken for four variables: Feb maximum daily mean 2-m temperature anomaly from ERA5-land; Feb maximum daily mean land surface skin temperature anomaly from ERA5-land; Feb maximum daily mean top layer soil temperature anomaly; and Feb 1-day maximum daily maximum temperature anomaly. The detailed methods for observational-based



attribution are used according to the World Weather Attribution Protocol, described in Philip et al. (2020), with supporting details found in van Oldenborgh et al. (2021), and Ciavarella et al. (2021). We also briefly outline the methods in Section

3.2.2 for the readers' interest. The results shown in Figure 11 suggest that the likelihood of events, in terms of Feb temperature extremes, at least as extreme as what happened in February 2023 is increased due to global warming, with a return period of 1.2/1.3/1.3/1.3 (for the four variables respectively) years under 2023 climate condition, a return period of ~11/20/22/9 years under -1.2 °C climate condition, and a probability ratio of ~9/15/17/7, i.e., the global warming has increased the likelihood of occurrence for Feb temperature anomaly at least as severe as the 2023 Feb over Rasac by almost

10-20 times, highlighting the impacts of climate change on local warming.

**Figure 11. The Generalized Extreme Value (GEV) fit to the data at two levels of the covariate global mean sea surface temperature: in 2023 (red line), and in a 1.2 degrees cooler climate (pre-industrial, blue line). The purple line shows the magnitude**

**of the February 2023 event. The four panels show results for four different evet definitions: (a) event defined based on Feb maximum daily mean 2-m temperature anomaly from ERA5-land; (b) event defined based on Feb maximum daily mean land surface skin temperature anomaly from ERA5-land; (c) event defined based on Feb maximum daily mean top layer soil temperature anomaly; and (d) event defined based on Feb 1-day maximum daily maximum temperature anomaly, from ERA5 (as ERA5-land doesn't provide this variable). Data over 1950-2023 is used, and all anomalies are with respect to 1980-2010**

**climatology period.**



## 5.4 Regional implications of the 2023 Rasac GLOF

Regional-scale insights about possibly changing occurrence patterns of GLOFs in space and time can be derived from systematic analysis of events with specified triggering events (type and magnitude) in homogenous geographical region. In this study, we consider 'large' (volume $>10^6$ m$^3$; using the universal landslide size classification scheme of McColl and Cook, 2024) ice and rock mass failure-triggered in glacierized Cordilleras of the Peruvian Andes. Building on regional the GLOF inventory of Emmer et al. (2022b), we found four examples of such GLOFs that are documented from the Peruvian Andes in 21$^{st}$ Century. These are the 2002 Lake Safuna Alta rockslide-induced GLOF in C. Blanca (Hubbard et al., 2005), the 2012 Lake No. 513 ice-rock avalanche-induced GLOF in C. Blanca (Carey et al., 2012), the 2020 Lake Salkantaycocha ice-rock avalanche-induced GLOF in C. Vilcabamba (Vilca et al., 2021) and the 2022 Lake Upiscocha rockslide-induced GLOF in C. Vilcanota (Vilca et al., 2022). The timing of these events implies that, while there was on average one large ice and rock mass failure-triggered GLOF per decade in the Peruvian Andes in 2000s and 2010s, this figure increased threefold in the 2020s. The rather rare nature of these events underscores the importance of careful reporting and analysis that is necessary to disentangle occurrence patterns and the role of climate change (see Section 5.2).

## 6 Conclusions

The GLOF process chain reported in this study is the most recent large mass movement (initial volume $>10^6$ m$^3$) triggered GLOF process chain in the Peruvian Andes. The triggering mass movement detached from cold, deep-reaching permafrost and the failure of the main event must have occurred within the frozen rocks. These frozen rocks have most likely undergone deep warming and related stability reduction but not to especially critical conditions of warm permafrost with increased amounts of unfrozen water, which is corroborated by the observed period of six years of precursory small magnitude rock fall activity. Our field observations emphasize the mobilization of very large boulders (5 boulders with diameter > 3 m) and surprisingly limited geomorphic impacts downstream the Lake Gochacotan. While hydrodynamic analysis reveals flow velocities exceeding 5 m/s for the translational motion and 10 m/s for the rotational motion, understanding the attenuation role of floodplain topography and the Lake Gochacotan requires detailed modelling. The 2023 Rasac GLOF in the context of similar events in the Peruvian Andes suggests that the frequency of such events has been increasing since the beginning of the 21$^{st}$ Century. We assume that this increase may primarily be attributed to climate change-induced ice effects, i.e., glacial de-buttressing and deep warming with stability reduction of perennially frozen rock slopes. These two dominant ice-related effects on changing preconditioning must be discussed in view of the challenges tied with attributing short-term triggering factors for individual events. In conclusion, we recommend that future research should especially address: (i) quantitative analysis of stability conditions of lake-facing slopes with probable permafrost conditions; (ii) conceptual and modelling-based analysis of amplification and attenuation elements in GLOF process chains; (iii) frequency-magnitude analysis of GLOF triggering events in changing climate.



**Acknowledgement:** The authors acknowledge the financial support (APC coverage) by the University of Graz.

**Code/Data availability:** The data analyzed in this study (remote sensing data, climate reanalysis data) are freely available
(see details in the main text). Any additional information will be provided by the corresponding author upon request.

**Author contributions:** All co-authors contributed to the discussion of research design, data compilation, formal analysis
and writing of the text. In particular, AE and SC analysed remotely sensed optical images and topographical data, OV and
CSC provided data from the archives and talked to eyewitness in the field in 2023, AE did the field visit in 2019, SL
processed and analysed climate data and wrote the attribution part, JH and EP analysed transport of large boulders and WH
examined thermal conditions and the role of warming permafrost. All co-authors approved the final version of this text.

**Competing interests:** None of the authors has any competing interests.

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
