# Peer review of "Causes, consequences and implications of the 2023 landslide-induced Lake Rasac GLOF, Cordillera Huayhuash, Peru"

_EGUsphere, 2024_

## Referee Comment (RC2)

egusphere-2024-2316
Causes, consequences and implications of the 2023 landslide-induced Lake Rasac
GLOF, Cordillera Huayhuasch, Peru

Emmer et al., 2024
* * *
**General comments**

The authors provide a description of a rock-avalanche induced GLOF from Lake Rasac in Peru and also discusses climate change attribution. The study provides some interesting information, but it somehow seems unfinished and I missed a clear take-away message. I have included a variety of specific comments below, but the main justification for me to investigate an event that had very little impact is to understand what attenuated the impact. This is currently done in passing, but without specific analysis. For example, it would be very interesting to know whether the second lake or a piece-wise fragmentation of the landslide was the key factor for the attenuation, a question that could be explored through some basic modeling. This would be an important takeaway for hazard assessments in other places.

The second question that the paper discusses is that of climate change attribution. This question is of utmost importance, and, as the authors discuss, is very hard (if not impossible) to answer for this setting. For me, the combination of the somewhat vague description and discussion of the actual event paired with the somewhat vague discussion of the climate change attribution leads to something that feels a bit like a collage. The attribution part would be more powerful if it included more locations/times (e.g., the dates of the other GLOFS), provided more context on absolute temperatures at different elevations etc. Understanding the return periods of temperature anomalies is interesting, but if the return period for a given warming event is nowadays ~1year, then there are many days/weeks during which no mass movements happen.

In addition to these two main points, the manuscript falls short on the discussion or presentation of many points. For example, statements like "significantly increased" often seem purely qualitative, when a quantification does not seem too hard. I have elaborated on these in the specific comments and a few technical corrections to the best of my abilities below.

**Specific comments**
L. 20: What do you mean by arête ridge? Seems redundant – would ridge not suffice?
L.20ff: The statement about the volume and origin from "zone with cold, deep permafrost" sounds like this information was pre-established. However, it sems that this information was in fact generated by this study. The abstract should reflect this, possibly saying a little bit more about how this information was procured.
L.28: Can you provide time scales for the "statistically significant" temperature rise and anomaly. It seems they could be very different?

L.30: can you be more specific about what made the geologic situation "already critical?"

L.41: Not hugely more insightful, but you could consider adding the reference to Bondesan, A., & Francese, R. G. (2023). The climate-driven disaster of the Marmolada Glacier (Italy). *Geomorphology*, *431*, 108687.

L.43: The term "piedmont areas" does not seem particularly well established (since you are not referring to the region in Italy). Consider simplifying.

L.66: Maybe add statement about why such "evidence-driven understanding" is important and who it can serve?

L.71: Maybe add elevation range?

L.83: Do you think the snow line elevation is still accurate for today's conditions? Maybe add a statement to qualify this?

L.85: Interesting discrepancy between the two studies that is a bit confusing at first, but probably are consequence of the image resolution. Maybe reformulate to state that there is a relatively robust estimate of lake area, but not so much about lake number (which maybe does not matter since the small ones don't really matter)? Could also compare to dataset from Shugar 2020?

L.88: Slightly confusing that a second lake is suddenly mentioned here.

L.90: what does it mean to characterize by "short longevity"? This whole paragraph seems a bit lost. Consider deleting/reformulating/refocusing. You don't need to mention anymore at this point of the paper that lake Rasac is the focus of the study.

L.109: I was expecting the description of a second lake here... I don't think it necessarily needs a separate section, but something about the distance between the lakes, the elevation difference etc. seems like it would be useful.

Section 2: I was maybe missing a little bit something about the general climate of the region (MAAT at the lake and the summit region, typical precipitation and weather patterns etc.)

L.132: Sentence spans 6 (!) rows of text... Consider merging the list of parameters that were derived from ERA-data to table 1?

L.145: I don't think the "event as defined above" has been defined above, or at least I don't understand what likelihood of occurrence has been assessed here.

L.147: What is temperature-GMST ?

P.3.2.2. I find the first paragraph very hard to follow. Can you provide a slightly more detailed description of what you did and maybe avoid the use of unnecessary acronyms (i.e., PR – it's not thaaaat long ;-) ).

S. 3.4: I am a bit surprised that the two most commonly used permafrost products by Gruber et al., or Obu et al., were not used for the permafrost assessment (Gruber, S. 2012: Derivation and analysis of a high-resolution estimate of global permafrost zonation, The Cryosphere, 6, 221-233. doi:10.5194/tc-6-221-2012; https://microsite.geo.uzh.ch/cryodata/pf_global/; Obu, J. (2021). How much of the earth's surface is underlain by permafrost?. *Journal of Geophysical Research: Earth Surface*, *126*(5), e2021JF006123; https://doi.pangaea.de/10.1594/PANGAEA.905512?format=html#download ). Can you elaborate on why the global datasets were not used. If the omission was an oversight, I think it would be useful to include them in the analysis.

L.220: Not entirely clear what "under these conditions" refers to. The last sentence of the paragraph above? The entire last paragraph? It almost seems that this entire

sentence could just be omitted. If some condition is important, it would be helpful to clarify.

S.4: Somewhere, either in the introduction, integrated into section 2 or latest at the beginning of section 4 (though that feels a bit late), a brief overview of the event would be very helpful. As a first time-reader, all I know at the point of starting on 4.1.1. is that there was a rock-failure triggered GLOF and I'm immediately presented with detailed facts about the ice aprons and hanging glaciers and the development of rockfall areas. I think some context about the general chain of events would be very useful up-front, maybe including a map of the runout zone and impacted area (which doesn't seem to appear anywhere else in the paper).

L.227: Can you provide some information about the bedding planes and their dip and strike and how these relate to the terrain slope? This would be more informative than "tilted layers". Also, rather than saying "right bank" maybe use cardinal directions?

L.239: How do you know the timing?

L.241: Is there truly no way to get a post-event DEM to better constrain the volume?

L.265: can you express the "substantial increase" in a quantitative way?

L.270: can you express the "rising significantly" in a quantitative way?

L.304: Maybe specify which Alps you mean? Based on the reference I take it that this is the European Alps? Maybe include some newer references for the statement on the aspect differences (e.g., Kenner et al., 2019 https://tc.copernicus.org/articles/13/1925/2019/ and Böckli et al., 2012 https://tc.copernicus.org/articles/6/807/2012/)

L.317: I think I miss a statement about why these horizontal gradients matter? What do they do? You state that "such thermal asymmetries are not uncommon in detachment zones with permafrost rock avalanches", but it's not clear why they matter. Ridges will frequently have strong heat-flux gradients and they are prone to mass movements (that's why they are ridges). Can you elaborate on the causal relationship?

S.4.2.2. Generally feels very discussion-y

S.4.2.3. This seems like an unreasonably short paragraph with very little information that I learn no conclusion from. Showing only 2 months of data to argue that the temperatures "started to become anomalously warm" is rather unsatisfactory, especially when a similar looking peak in January is obscured by the legend. I would be nicer to see these data in the longer term context (how "anomalous" is this?) and also learn what the absolute temperatures were. How much warmer than usual were things? Fig. 8 also includes precipitation, but this is not discussed at all. If it doesn't show anything (which it appears not to, just don't show it).

S.4.3 I think I would move this section up before the met-analysis. It provides more context for what happened.

L.361: can you quantify the "pronounced steepness"?

L.363: it would be very nice to see some of the photos taken of the site post-event!

L.360: was there any change in the height of the moraine dam? This seems like it would be a very important insight for future hazards, since may limit the possible lake level.

L.385: "this may explain the two dam overtopping locations" → I think this is the first I read of this, but it should probably be described somewhere in the results.

L.392: This Salkantaycocha GLOF is mentioned twice here in the discussion, but has not been introduced. Given its similarity but different outcome, I think introducing the reader to it in the introduction would be warranted.

L.395: I am a bit confused by the deposition of the frontal part vs. clear water apparently having been seen in the fontal part of the lake after the GLOF. I think (similar to comment above on S.4), some simple mapping would be nice to clarify the different areas.

S. 5.3: For me, the results of the attribution analysis belong in the results section, not in the discussion.

L.430: "A promising way forward..." This statement is interesting but it is not at all backed up or provided in context. Has this been done? (→ references)? If not, why not? (→context).

L.433: "overall trend is clear" → references?

F.11: I do not get the reference to "global mean sea level temperature". Is this supposed to be in the caption?

S.5.4: Feels unfinished and not very insightful.

L.470: These are very few events to be reporting averages and increases. Is there a chance that certain events could have been missed or do you exclude that? You could include a statement about this. I also wondered whether the size classification is the right way by which to select events. If a smaller triggering event led to a destructive (or not – since that does not seem to be the criteria) GLOF, then it seems like it would also qualify?

**Technical corrections**

Personally, I find that things that were done by the authors should be described in the past tense. E.g., we analyzed, we evaluated etc.

L. 61: wording makes it sound like there were numerous GLOFs from Lake Rasac and the authors are only examining the most recent one. I don't understand this to be the case, so the wording should be adjusted (if this is wrong, then background information on prior events should be provided.)

L.70: north *of* Lima

L.82: *the* (total?) glacier extent

L.83: Suggest to move reference (McFadden) to the end of the sentence.

Fig. 1: I'm really perplexed by the pink glaciers ;-) Can you just make them white? In (C) can you point out the important lakes, it took a bit of staring to find it. Generally, I think the colors could be improved (white ocean, yellow mountains, missing coastline etc.)

Fig. 2: Recurring rockfall release zone is very hard to read. What does images AE mean? Please provide image dates. (C) these are all oblique views, so there is no need to specify this for the last image. I'm still confused what an "arête ridge" is. If this is important, please define it.

L.106: insert space lakeand

L.115: "analyzed" instead of "integrated the analysis of"

L.116: "characterize" or "describe" rather than "estimate spatial characteristics of"

L.149: superfluous - after climate

L.266: "solid rainfall, i.e. snowfall" → just write snowfall ;-)

L.360: "a moraine dam" → is there more than one?

L.421: "human-induced increased energy content of the climate system" seems a very roundabout way of saying CO2 increase. Could be stated in more simple terms.

L.429: ...attribution of event is challenging: This feels fairly repetitive at this point in the section.

L.455: "evet"

L.465: There is a discrepancy in the year of McColl & Cook between in text reference (2024) and the bibliography (2023)

---

## Author Comment (AC1)

We thank the two reviewers for their detailed comments and suggestions to our manuscript *Causes, consequences and implications of the 2023 landslide-induced Lake Rasac GLOF, Cordillera Huayhuash, Peru*. We have considered them in revised version of the manuscript. Below we provide our point-by-point replies (in blue).

On behalf of the collective of authors
Adam Emmer

- - -

**RC2**: 'Comment on egusphere-2024-2316', Anonymous Referee #2, 14 Oct 2024

egusphere-2024-2316 Causes, consequences and implications of the 2023 landslide-induced Lake Rasac GLOF, Cordillera Huayhuasch, Peru Emmer et al., 2024 --------------------------------------------------- --------------------------------------------------------
General comments
The authors provide a description of a rock-avalanche induced GLOF from Lake Rasac in Peru and also discusses climate change attribution. The study provides some interesting information, but it somehow seems unfinished and I missed a clear takeaway message. I have included a variety of specific comments below, but the main justification for me to investigate an event that had very little impact is to understand what attenuated the impact. This is currently done in passing, but without specific analysis. For example, it would be very interesting to know whether the second lake or a piece-wise fragmentation of the landslide was the key factor for the attenuation, a question that could be explored through some basic modeling. This would be an important takeaway for hazard assessments in other places.

- We are convinced there are number of reasons why such events should be researched (understanding GLOF occurrence in space and time; understanding GLOF process chains) and that these outcomes may be of interests for both scientific community (high mountain geomorphology) as well as DRR practitioners (natural hazard science)
- Our study primarily described the 2023 GLOF, elaborates on the triggers, characteristics of a GLOF process chain and discuss the role of changing climate change
- We document two rather contradicting observations: (i) transport of very large boulders (diameter > 3m) by the GLOF; and (ii) substantial attenuation of the flood in the downstream located lake
- As such, the findings presented in this case study could rather be used to challenge / validate existing models, instead of models used to analyze the event; this is, however, beyond the scope of the study

The second question that the paper discusses is that of climate change attribution. This question is of utmost importance, and, as the authors discuss, is very hard (if not impossible) to answer for this setting. For me, the combination of the somewhat vague description and discussion of the actual event paired with the somewhat vague discussion of the climate change attribution leads to something that feels a bit like a collage. The attribution part would be more powerful if it included more locations/times (e.g., the dates of the other GLOFS), provided more context on absolute temperatures at different elevations etc. Understanding the return periods of temperature anomalies is interesting, but if the return period for a given warming event is nowadays ~1year, then there are many days/weeks during which no mass movements happen.

- As discussed in the original manuscript, and as pointed out by the reviewer, carrying out climate change attribution for GLOF events is very hard (if not possible), given the current challenges as discussed in the original manuscript, which we briefly elaborate further here: 1) first and foremost, we need to identify the causal physical process pathway linking the GLOF event to weather/climate variables (con-current with the GLOF events, and/or antecedent conditions) that triggers the GLOF or made the GLOF more susceptible; in the case of Rasac, this could either be exploring the immediate weather condition to establish whether they serve as a trigger, or be investigating whether the antecedent weather/climate conditions contributed to the mass movement processes. The lack of clear understanding and targeted modelling efforts to unpack the role played by weather/climate in the mass movement processes→GLOF is the first big challenge. 2) To perform a robust analysis of climate change attrition of GLOF events, the weather/climate related drivers of GLOF need to be identified (as discussed in (1)), and this includes identifying the responsible meteorological variables, their geographic and temporal scales (whether it's the weather conditions concurrent with the GLOF or the antecedent conditions), the start and end data of the weather/climate period/season that are of interest, and the appropriate counterfactual scenario (in which the effect of climate change is excluded). For earth surface surface processes such as landslides and GLOFs, defining the counterfactual scenario involves defining how the earth surface and glacial lake evolution processes unfold without the effect of climate change. These are the reasons why in the original manuscript we made the statements that 'A detailed and systematic attribution of glacial lake expansion and GLOFs to global warming has not yet been undertaken and the roadblocks to such efforts have been a lack of clear understanding of glacial lake evolution processes and a lack of modelling of such processes… A promising way forward to attributing GLOFs is to use integrated process chain glacier-climate modelling to address if and by how much glacial lake expansion and GLOFs are attributable to past total global warming and to the anthropogenic component.'
- Nonetheless, we agree with the reviewer that climate change attribution of specific GLOF events is extremely important, and we start to explore some initial steps to perform such analysis.
   Following the reviewer's advice, we have added attribution analysis on two addition recent events of similar nature in this area.

| Lake | Cordillera | GLOF date | LONG | LAT | LAKE ELEV | MASS MOVEMENT |
|---|---|---|---|---|---|---|
| Salkantaycocha | Vilcabamba | 23rd February 2020 | -72.569411 | -13.342400 | 4470 | ice-rock avalanche |
| Upiscocha | Vilcanota | 9th August 2022 | -71.261018 | -13.770244 | 4558 | rock slide / avalanche |
| Rasac | Huayhuash | 12th February 2023 | -76.938047 | -10.264378 | 4654 | ice-rock avalanche |

In addition to these two main points, the manuscript falls short on the discussion or presentation of many points. For example, statements like "significantly increased" often seem purely qualitative, when a quantification does not seem too hard. I have elaborated on these in the specific comments and a few technical corrections to the best of my abilities below.
   - We have quantified "significant increase" where possible

Specific comments

L. 20: What do you mean by arête ridge? Seems redundant – would ridge not suffice?

- the word "arête" highlights very steep slope, knife-edge ridge morphology and we prefer to keep using this term in our study

L.20ff: The statement about the volume and origin from "zone with cold, deep permafrost" sounds like this information was pre-established. However, it sems that this information was in fact generated by this study. The abstract should reflect this, possibly saying a little bit more about how this information was procured.

- True; we reformulate as follows: " … triggered by a mass movement from the failure of an arête ridge with an estimated volume of 1.1 to 1.5 x 106 m3; this occurred *from a rock zone where climate information – primarily from reanalysis data – indicates cold*, deep permafrost, and was preceded by several small-magnitude precursory rockfall events."

L.28: Can you provide time scales for the "statistically significant" temperature rise and anomaly. It seems they could be very different?

- Thanks for pointing this out. We have removed 'statistically significant' from L28, becasue through the analysis what we can establish is the trend over the last 8 decades (1940-2023) as described in the first half of that sentence, but the second half of the sentence is referring to the weather conditions immediately prior to the GLOF event.

L.30: can you be more specific about what made the geologic situation "already critical?"

- this sentence was reformulated

L.41: Not hugely more insightful, but you could consider adding the reference to Bondesan, A., & Francese, R. G. (2023). The climate-driven disaster of the Marmolada Glacier (Italy). Geomorphology, 431, 108687.

- Suggested reference has been added

L.43: The term "piedmont areas" does not seem particularly well established (since you are not referring to the region in Italy). Consider simplifying.

- Revised accordingly ("foothill" instead of "piedmont")

L.66: Maybe add statement about why such "evidence-driven understanding" is important and who it can serve?

- A justification statement "Such outcomes can serve scientific community (in particular geomorphologists, natural hazard scientists and process chain model developers) as well as disaster risk reduction authorities and practitioners." has been added

L.71: Maybe add elevation range?

- The elevation of the highest peak (Nev. Yerupajá) is mentioned

L.83: Do you think the snow line elevation is still accurate for today's conditions? Maybe add a statement to qualify this?

- Revised accordingly

L.85: Interesting discrepancy between the two studies that is a bit confusing at first, but probably are consequence of the image resolution. Maybe reformulate to state that there is a relatively robust

estimate of lake area, but not so much about lake number (which maybe does not matter since the small ones don't really matter)? Could also compare to dataset from Shugar 2020?

- Revised accordingly; comparing with the global dataset of Shugar et al., 2020 doesn't make a lot of sense as they only include lakes > 50,000 m2 and there are only 11 lakes exceeding this threshold in the C. Huayhuash

L.88: Slightly confusing that a second lake is suddenly mentioned here.

- Revised accordingly

L.90: what does it mean to characterize by "short longevity"? This whole paragraph seems a bit lost. Consider deleting/reformulating/refocusing. You don't need to mention anymore at this point of the paper that lake Rasac is the focus of the study.

- Agree, the part of this has been moved to Intro and the paragraph has been deleted

L.109: I was expecting the description of a second lake here… I don't think it necessarily needs a separate section, but something about the distance between the lakes, the elevation difference etc. seems like it would be useful.

- Revised accordingly

Section 2: I was maybe missing a little bit something about the general climate of the region (MAAT at the lake and the summit region, typical precipitation and weather patterns etc.)

- Thanks for the suggestion, we have added a few sentences on the general climate of the region.

L.132: Sentence spans 6 (!) rows of text… Consider merging the list of parameters that were derived from ERA-data to table 1?

- Thanks for the suggestion, we have added the list of meteorological variables from ERA5 to the new table

L.145: I don't think the "event as defined above" has been defined above, or at least I don't understand what likelihood of occurrence has been assessed here.

- deleted

L.147: What is temperature-GMST ?

- global mean surface temperature (GMST)

P.3.2.2. I find the first paragraph very hard to follow. Can you provide a slightly more detailed description of what you did and maybe avoid the use of unnecessary acronyms (i.e., PR – it's not thaaaat long ;-) ).

- Revised accordingly

S. 3.4: I am a bit surprised that the two most commonly used permafrost products by Gruber et al., or Obu et al., were not used for the permafrost assessment (Gruber, S. 2012: Derivation and analysis of a high-resolution estimate of global permafrost zonation, The Cryosphere, 6, 221-233. doi:10.5194/tc-6-221-2012; https://microsite.geo.uzh.ch/cryodata/pf_global/; Obu, J. (2021). How much of the earth's surface is underlain by permafrost?. Journal of Geophysical Research: Earth Surface, 126(5),

e2021JF006123; https://doi.pangaea.de/10.1594/PANGAEA.905512?format=html#download ). Can you elaborate on why the global datasets were not used. If the omission was an oversight, I think it would be useful to include them in the analysis.

- The two mentioned sources of large-scale climate-based model results are indeed important and are now cited (see below). Because of their low spatial resolution (about 1 km), however, they only provide general indications (cf. example from the Gruber simulation):

[Figure]

- we now reformulate in the following way: "The assessment of permafrost conditions is based on the analysis of Google Earth Images, (sparse) climatic information, experience of borehole temperatures available in Europe, and scientific literature about slope stability of ice-clad and perennially frozen mountain peaks. They constitute "best estimates" for a site without direct measurements, dealing with thermal conditions, subsurface and surface ice properties, and resulting hydro-mechanical aspects in view of climate-related stability changes. Large-scale (global) model results relating to patterns of permafrost occurrence are available from Gruber (2012) based on climate data for the second half of the 20$^{th}$ century (1960-1990) and from Obu et al. (2021) based on climate data (freezing/thawing indices) for the beginning of the 21$^{st}$ century (2000-2016) in combination with empirical thermal offsets mainly related to effects from vegetation and winter snow cover; comparison with measured borehole temperatures indicate an uncertainty range of $\pm$ 2°C. Due to the low spatial resolution (about 1 km) of these two important approaches, topo- and microclimatic effects at the local Rasac detachment site in an extremely steep slope must be assessed using additional process-related indications as described further below. "
- The references: Gruber, S. 2012: Derivation and analysis of a high-resolution estimate of global permafrost zonation, The Cryosphere, 6, 221-233. doi:10.5194/tc-6-221-2012; https://microsite.geo.uzh.ch/cryodata/pf_global/
  Obu, J. (2021). How much of the earth's surface is underlain by permafrost?. Journal of Geophysical Research: Earth Surface, 126(5), e2021JF006123; https://doi.pangaea.de/10.1594/PANGAEA.905512?format=html#download ).
  are now included in the reference list.

L.220: Not entirely clear what "under these conditions" refers to. The last sentence of the paragraph above? The entire last paragraph? It almost seems that this entire sentence could just be omitted. If some condition is important, it would be helpful to clarify.

- This sentence has been removed

S.4: Somewhere, either in the introduction, integrated into section 2 or latest at the beginning of section 4 (though that feels a bit late), a brief overview of the event would be very helpful. As a first time-reader, all I know at the point of starting on 4.1.1. is that there was a rock-failure triggered GLOF and I'm immediately presented with detailed facts about the ice aprons and hanging glaciers and the development of rockfall areas. I think some context about the general chain of events would be very useful up-front, maybe including a map of the runout zone and impacted area (which doesn't seem to appear anywhere else in the paper).
- After thorough discussion within the team of the co-authors, we came to the conclusions that the structure of the paper makes sense as it is, although we understand the reason why the reviewer made this comment

L.227: Can you provide some information about the bedding planes and their dip and strike and how these relate to the terrain slope? This would be more informative than "tilted layers". Also, rather than saying "right bank" maybe use cardinal directions?
- the sentence has been reformulated, however, the requested info about bedding planes, dip and strike are not available since the majority the upper part of the ridge is covered by ice; detailed look a the release zone shown in Fig. 3A reveals distinguishable layers that are inclined parallel with the slope

L.239: How do you know the timing?
- we estimated time window of the GLOF occurrence from remote sensing images (several days) which was narrowed down by the members of the community settled in the Jahuacocha valley

L.241: Is there truly no way to get a post-event DEM to better constrain the volume?
- perhaps a drone survey could provide better volume estimates, it is, however restricted by two reasons: (i) elevation and accessibility; (ii) nature protection interests; our resources do not allow it
- further, existing pre-event DEMs (SRTEM DEM and ALOS PALSAR) suffer from voids and associated interpolations

L.265: can you express the "substantial increase" in a quantitative way?
- these numbers are shown in corresponding figures extended back to 1940s; we also mention them in the revised version of the text

L.270: can you express the "rising significantly" in a quantitative way?
- these numbers are shown in corresponding figures extended back to 1940s; we also mention them in the revised version of the text

L.304: Maybe specify which Alps you mean? Based on the reference I take it that this is the European Alps? Maybe include some newer references for the statement on the aspect differences (e.g., Kenner et al., 2019 https://tc.copernicus.org/articles/13/1925/2019/ and Böckli et al., 2012 https://tc.copernicus.org/articles/6/807/2012/)
- Thanks for the suggestion. The formulation has been extended to: In the *European* Alps, the difference between permafrost occurrence on E- and W-oriented slopes is around 500 m *as*

*already noted* by Haeberli (1975) and *implemented by* Keller 1992; *cf. later model approaches by Boeckli et al, 2012 or Kenner et al. 2019*).

- The references:
- Boeckli, L., Brenning, A., Gruber, S., Nötzli, J. 2012. Permafrost distribution in the European Alps: calculation and evaluation of an index map and summary statistics. The Cryosphere 6: 807-820. https://tc.copernicus.org/articles/6/807/2012/).
- Kenner, R., Noetzli, J., Hoelzle, M., Raetzo, H., and Phillips, M.: Distinguishing ice-rich and ice-poor permafrost to map ground temperatures and ground ice occurrence in the Swiss Alps, The Cryosphere, 13, 1925–1941, https://doi.org/10.5194/tc-13-1925- 2019, 2019.
- Have been added to the reference list.

L.317: I think I miss a statement about why these horizontal gradients matter? What do they do? You state that "such thermal asymmetries are not uncommon in detachment zones with permafrost rock avalanches", but it's not clear why they matter. Ridges will frequently have strong heat-flux gradients and they are prone to mass movements (that's why they are ridges). Can you elaborate on the causal relationship?

- The main reason for mentioning such asymmetries primarily is to correctly mention them. We however added the following brief statement: "In the uppermost part of the Rasac detachment, where the ridge width is around 100 – 200 m, temperature gradients (around 1 - 3°C/100 m) indicate a horizontal heat flow component, which is comparable to the geothermal heat flow in flat topography under conditions of thermal equilibrium. *Such horizontal asymmetries and related, relatively weak heat fluxes nevertheless have the potential to become critical for slope stability in connection with warming-induced water infiltration from the warm side of ridges.*"

S.4.2.2. Generally feels very discussion-y

- We agree this part may feel discussion-y, however, after careful discussion and considering overall structure of the manuscript, we prefer to keep it where it is

S.4.2.3. This seems like an unreasonably short paragraph with very little information that I learn no conclusion from. Showing only 2 months of data to argue that the temperatures "started to become anomalously warm" is rather unsatisfactory, especially when a similar looking peak in January is obscured by the legend. I would be nicer to see these data in the longer term context (how "anomalous" is this?) and also learn what the absolute temperatures were. How much warmer than usual were things? Fig. 8 also includes precipitation, but this is not discussed at all. If it doesn't show anything (which it appears not to, just don't show it).

- Thanks for the suggestion, we have added in two additional panels in figure 8, to show the climate conditions for 2 years prior (2021 and 2022). We plotted everything in anomaly terms, to show how much warmer things are than the usual conditions. The baseline usual conditions were represented by a 30-year mean climatology over 1981-2010, as recommended by the World Meteorological Organization. The anomaly is calculated with respect to each month, i.e., Jan 2023 anomaly is calculated by subtracting the Jan climatology from the Jan 2023 absolute values, to take away the influence of seasonality. We still think it's more valuable to show the anomaly, instead of absolute temperatures, because it is how anomalous the conditions are compared to usual, that are important in extreme events. Although

precipitation didn't show much, we still think it's valuable to show precipitation here, in order to establish that this event is not caused by sudden high precipitation accumulation.

S.4.3 I think I would move this section up before the met-analysis. It provides more context for what happened.
- We decided not to change the structure of the paper

L.361: can you quantify the "pronounced steepness"?
- revised accordingly

L.363: it would be very nice to see some of the photos taken of the site post-event!
- We added few post-event field images to which we get the permission from the author

L.360: was there any change in the height of the moraine dam? This seems like it would be a very important insight for future hazards, since may limit the possible lake level.
- the dam was eroded during the overtopping (not breached or incised); since the lake doesn't exist anymore, we do not elaborate on future GLOF hazard

L.385: "this may explain the two dam overtopping locations" → I think this is the first I read of this, but it should probably be described somewhere in the results.
- this is described in Section 4.3.1 GLOF process chain

L.392: This Salkantaycocha GLOF is mentioned twice here in the discussion, but has not been introduced. Given its similarity but different outcome, I think introducing the reader to it in the introduction would be warranted.
- the Salkantaycocha GLOF has been mentioned in the Intro section

L.395: I am a bit confused by the deposition of the frontal part vs. clear water apparently having been seen in the fontal part of the lake after the GLOF. I think (similar to comment above on S.4), some simple mapping would be nice to clarify the different areas.
- new Fig. 11 has been added

S. 5.3: For me, the results of the attribution analysis belong in the results section, not in the discussion.
- Thanks for raising this. As discussed in response to the reviewer's second major point above, given we still can't perform a straightly speaking robust attribution where we can establish the role played by weather/climate in the causal physical process pathway, we still think briefly discussing what the attribution would look like if we were to just focus on the temperature anomaly is useful to have in the discussion, in the same section where we discuss the challenges and ways forward.

L.430: "A promising way forward…" This statement is interesting but it is not at all backed up or provided in context. Has this been done? (→ references)? If not, why not? (→context).
- Thanks for raising this, hopefully our response to the reviewer's second major point has further clarified why this hasn't been done, and we have revised the text to provide more context in the revised manuscript.

L.433: "overall trend is clear" → references?
- references supporting this statement have been added

F.11: I do not get the reference to "global mean sea level temperature". Is this supposed to be in the caption?
- Apologies for the confusion, the figure caption should have referred to global mean surface temperature, instead of global mean sea surface temperature. The global mean surface temperature was referring to the attribution method (L147 in the original manuscript) as detailed in section 3.2.2. This has been corrected.

S.5.4: Feels unfinished and not very insightful.
- this section has been reworked and extended

L.470: These are very few events to be reporting averages and increases. Is there a chance that certain events could have been missed or do you exclude that? You could include a statement about this. I also wondered whether the size classification is the right way by which to select events. If a smaller triggering event led to a destructive (or not – since that does not seem to be the criteria) GLOF, then it seems like it would also qualify?
- This part will be reformulated (we simply report the basic numbers - that there were GLOFs triggered by large landslides in 2002, 2012, 2020, 2022 and now 2023, and that, whilst this is a small set of datapoints, it is noteworthy (and I choose that word carefully) that more of those have occurred in the last 4 years than in the preceding 2 decades); and our observations will be put in the context of the insights from other regions
- beyond the scope of this study, we aim to tackle the probabilities of GLOF occurrence in a quantitative way and we find it more relevant to look at size classification (and frequency-magnitude relationships) of triggers rather than "magnitude" or resulting GLOFs which can be attenuated / amplified by various factors / processes / settings

Technical corrections
Personally, I find that things that were done by the authors should be described in the past tense. E.g., we analyzed, we evaluated etc.
- revised accordingly

L. 61: wording makes it sound like there were numerous GLOFs from Lake Rasac and the authors are only examining the most recent one. I don't understand this to be the case, so the wording should be adjusted (if this is wrong, then background information on prior events should be provided.)
- revised accordingly

L.70: north of Lima
- revised accordingly

L.82: the (total?) glacier extent
- revised accordingly

L.83: Suggest to move reference (McFadden) to the end of the sentence.
- revised accordingly

Fig. 1: I'm really perplexed by the pink glaciers ;-) Can you just make them white? In (C) can you point out the important lakes, it took a bit of staring to find it. Generally, I think the colors could be improved (white ocean, yellow mountains, missing coastline etc.)

- this Figure was revised, also in line with comments of Referee #1

Fig. 2: Recurring rockfall release zone is very hard to read. What does images AE mean? Please provide image dates. (C) these are all oblique views, so there is no need to specify this for the last image. I'm still confused what an "arête ridge" is. If this is important, please define it.

- Figure caption has been revised; AE (=Adam Emmer) is the author of images
- "Recurring rockfall release zone" text has been moved

L.106: insert space lakeand

- revised accordingly

L.115: "analyzed" instead of "integrated the analysis of"

- revised accordingly

L.116: "characterize" or "describe" rather than "estimate spatial characteristics of"

- revised accordingly

L.149: superfluous - after climate

- revised accordingly

L.266: "solid rainfall, i.e. snowfall" → just write snowfall ;-)

- revised accordingly

L.360: "a moraine dam" → is there more than one?

- revised accordingly

L.421: "human-induced increased energy content of the climate system" seems a very roundabout way of saying CO2 increase. Could be stated in more simple terms.

- revised accordingly

L.429: …attribution of event is challenging: This feels fairly repetitive at this point in the section.

- This sentence has been deleted

L.455: "evet"

- revised accordingly

L.465: There is a discrepancy in the year of McColl & Cook between in text reference (2024) and the bibliography (2023)

- unified to 2024

Thank you again for your review!

---

## Author Comment (AC2)

We thank the two reviewers for their detailed comments and suggestions to our manuscript *Causes, consequences and implications of the 2023 landslide-induced Lake Rasac GLOF, Cordillera Huayhuash, Peru*. We have considered them in revised version of the manuscript. Below we provide our point-by-point replies (in blue).

On behalf of the collective of authors
Adam Emmer

- - -

**RC1**: 'Comment on egusphere-2024-2316', Fabrizio Troilo, 24 Sep 2024

General Comment:

The paper by Adam Emmer et Al. presents the description and the analysis of a major mass movement event involving cascading processes that happened in February 2023 in the Peruvian Andes.

The paper is well written and well structured; the text is supported by high quality figures and the references cited highlight a good literature review.

The topic of GLOFs, cascading processes triggering GLOFs, and the evolution of such events in the present climate change scenarios are relevant.

I believe that the prompt data and informations gathering  after such large events is crucial for the better understanding of these processes which are still difficult to model and to predict. The rapid publishing of such observations and studies is essential for the scientific community that studies such processes.

The hydrodynamical analysis performed in the study is interesting and can serve as a basis for future, more in-depth modeling work on this event. Another relevant point is the understanding of features or conditions that can reduce the propagation of gravitational phenomena (section 5.1) which is crucial to be correctly interpreted and integrated in dynamic gravitational modeling and eventually in risk assessment studies.

The discussion of the attribution of the event to climate change and the analysis of the variation in the likelihood of the occurrence of extreme temperature anomalies is particularly interesting, and could be the subject of future work on this and other case studies.

I believe this is an interesting paper and I hereby introduce some specific comments which i consider all as minor revisions and suggestions that could nonetheless further improve an already very good work.

- thank you for overall positive evaluation of the work done and its relevance for developing GLOF research field

Specific comments:

L20 it might be better to express the value with the estimated uncertainity. 1.3 +/- 0.2 or so. Also Check and homogeneize with L383 and L387 notation.

- the volume estimate 1.1 to 1.5 · $10^6$ m$^3$ refers to estimated thickness of displaced block 30 to 40 m; we find this expression appropriate as 1.3 +/- 0.2 would suggest this estimate came from frequency distribution (which does not); we homogenized the way how this is written throughout the manuscript

L31 to make it more clear and fluent i would slightly rephrase: (ii) frequency-magnitude relationships of extreme geomorphic processes that undergo alteration because of rapidly changing ...

- revised accordingly

L33 the wording of the phrase is a bit complicate...because glacial debuttressing is implicitly an ice related effect, why dont contract into : GLOFs originating in recent decades from glacial de-buttressing and warming permafrost (areas?)

- revised accordingly ("... originating from warming cryosphere ...")

L62 just remove "then", "if so, how this event can be attributed" is clear enough

- revised accordingly

Fig1 the Cordillera Huayhuash outline can hardly be seen. Why don't indicate it in black in the legend and just stick to the rectangular bounding box? Then you can give the more accurate perimeter in (b). It could be a good idea to change outline color in (b) to make it more visible. I would also write the full C. HUAYHUASH in legend (b) C.H. could be not really intuitive.

- This figure will be revised also in line with comments of Referee #2

L103 Can you give a value of uncertainty for the volume estimation?

- The lake volume estimate method developed by Muñoz et al. does not quantify the uncertainty range, however, it is the most suitable method considering it is based on a large dataset of bathymetries of Peruvian glacial lakes

Table1 Could you add a column that refers to the native satellite or sensor type? For example on the first row it could be: mosaicking of worldview and Pléiades imagery...For the 2nd row Planetscope satellite imagery etc... In the reference column you could add a detail if freely available or of restrained/commercial access. For the Sentinel images it could be good to add a reference on how to access original imagery from The copernicus data space. A column with the number of images actually used in the study could make it more complete

- thanks for the suggestion, but we have decided to keep Table 1 as it is. It presents the fundamental information needed and is consistent with reporting in other papers. These are fairly standard datasets and most readers will likely know where to access imagery from in all cases, and their cost vs. open access availability

L146 I would put two commas: "the event, as defined above, has changed ... "

- revised accordingly

L202 It would be good to introduce the Ravanel 2023 reference here as well, just after mentioning the ice aprons, as many geoscientists might not be aware of the definition of ice aprons.

- revised accordingly

L223 Here you name it Rasac Ridge. Elsewhere in the paper you also refer to the Rasac Arete Ridge. It should be named in the same way in the whole text. Wouldn't it be more straightforward to name it Rasac Ridge everywhere in the text? (But I might be missing something about local topography naming)
- the ridge doesn't have any official name, we name it according to the highest peak; the word "arête" is used to underscore its morphology; we homogenized the naming throughout the text

Figure 3 It would be really interesting for the reader to see the slope on the other side of the ridge as well if an image is available. From this side of the slope it is hard to tell that the morphology of the ridge is the one highlighted in the topographic profile in Figure 5.
- this is exactly the reason why we used the term "arête" which describes this ridge morphology

L243 Just a typo, the reference appears in red.
- revised

Figure 4 The north arrow is quite small, as long as you are showing images with the north pointing to the left (which is good for the composition of the panels of the image sequence), I would highlight it more clearly for the reader; I would add a bigger north arrow on the august 2017 panel for example, or make it bigger and/or colored to differenciate it from the main legend elements in the actual location.
- Revised version of this figure will include larger north arrow

L256 Permafrost thawing might be more appropriate?
- Replaced by "permafrost warming" since we do not assume thawing (see Fig. 7)

L315 In this paraghraph you highlight the fact that the Rasac Ridge is very sharp and horizontal heat flow component is present. Because of this setting, wouldn't it be reasonable to discuss (maybe in 5.2) the possibility that glacial debuttressing from the elevation loss of the big glacier tongue located to the East of the Rasac ridge could have played a role in the destabilization process? Maybe the recent exposure of part of the eastern flank of the Rasac ridge have also had consequences in the variation of the thermal condition of the slope as well as water percolation od circulation? Glacial de-buttressing effects are introduced generally in the paper and it could be worth to contextualize it to the local setting. The data from the paper of (Hugonnet at Al. 2021) could maybe give an idea of the elevation loss rates on this Glacier. Maybe historical data or geomorphologic evidence can at least give an idea of the LIA glacier extentions in the area?
- Thanks for opening this topic; the vertical difference between the Rasac arete ridge and the glacier in the valley on the E side of the ridge is > 500 m; this suggests that it has been remodelled by repeated glaciations and it is very unlikely that the LIA ice thickness would come even close to upper parts of the ridge where the 2023 mass movement was initiated; and it is even more (>1000 m) on the W-facing side of the ridge; considering the location of the release zone very high up above the valley floor and glaciers, the substantial role of debuttressing in what has happened is considered unlikely in this setting

- Glacier debuttressing in the paper is mentioned in discussion and conclusion sections where we put the Rasac event in the context of other events from the Peruvian Andes in some of which it was an important process

L324 "interested by the detachment" instead of "with the detachment" probably sounds better to the reader.
- The sentence was revised

L325 Temperature rise "is" from both sides of the ridge. It might sound better: temperature rise "have origin" or "comes" .
- Revised accordingly

L330 In 4.2.3, the content of figure 8 is very clear, but the text is very short. You could perhaps describe this data a little more, for example: skin temperature reached the highest anomaly peaking at +1.4°C etc… the temperature anomaly started from mid-january and returned to less extreme values by end of February etc etc …
- Thanks for the suggestion, we have revised the text to provide a more detailed description of the updated figure 8, for which we have extended the data back to 1940 (as the data had become available recently) to provide a longer view of the climate conditions & trends.

A more general consideration on the analysis of meteorological conditions is the following:
You show that a strong positive temperature anomaly was present at the time of failure of the rock avalanche of 12th of February, which is really interesting, but wouldn'it be interesting to show a bit more data from the previous months? Or maybe from the previous 2 or 3 years? It seems that the reader would need a bit larger temporal outlook on these data. Maybe another 2 panels in figure 8 could highlight a little larger time window? In this direction you could also highlight in colored lines years 2021 and 2022 for example in the panels A, C and E of figure 6.
- Thanks for the suggestions. We have added 2 panels in figure 8, to show the meteorological conditions 2 years prior, showing the anomaly with respect to a 30-year climatology (1981-2010, same as what's been show in figure 8); we have also highlighted years 2021 and 2022 in the updated figure 6.

Figure 9 i would reproduce the (c) outline in (A) also, to better compare A and B panels
In (B) the label "GLOF impact area" covers the area itself, if you shift it to the north, the reader can better appreciate the change in the impact area occurred in between (A) and (B) panels
It could be good to have another panel highlighting the (D) area showing the same detail of this area prior to the GLOF. Remember to add a North arrow
- This figure will be revised accordingly

L364 "apparently"
- Revised accordingly

L366 isn't it referring to fig 9?
- Revised accordingly

L385 add a reference to fig 9. maybe it is also worth better highlighting the fact that there are 2 different overtopping locations on fig 9 with 2 additional circles or better separate the 2 arrows

- Revised accordingly

L387 don't express "millions" but 0.8 x $10^6$ m$^3$
- Revised accordingly

L389 it is not clear .. lake rasac persisted in a more limited extent .. compared to pre-glof extent?or refers to temporal extent?
- Clarified in the text

L393 just say unusual if it is so, dont say quite. You should also explain why it is unusual.
- Revised accordingly

L397 why is it rather unlikely?
- This is considered unlikely because deposition have not occurred in the rear part of the lake where it could possible interrupt water inflow into the lake, but a frontal part

L403 it would be more fluent by merging the two sentences: a partial role, in the sense ...
This way you eliminate the repetition of the word partial.
- Revised accordingly

L418 "prior to" sounds better than "leading up" if I interpreted correctly the meaning.
- Revised accordingly

Figure 11 Orange and purple coloring might be hard to differenciate.
(a) and (c) panels have the x axis label truncated on the 10000 label
- Revised accordingly

L456 Typo: event , not evet
- Revised accordingly

L465 Building on the regional
- Revised accordingly

L474 Maybe a little table with some BASIC data on the events you mention could make the paper more complete at this point.
- New Table 2 with basic description of the mentioned events has been added

Hoping that my comments will be useful for the publication of your manuscript,
Best regards,
Fabrizio

Thank you again for your review!